# Development and Comparison of Dengue Vulnerability Indices Using GIS-Based Multi-Criteria Decision Analysis in Lao PDR and Thailand

**DOI:** 10.3390/ijerph18179421

**Published:** 2021-09-06

**Authors:** Sumaira Zafar, Oleg Shipin, Richard E. Paul, Joacim Rocklöv, Ubydul Haque, Md. Siddikur Rahman, Mayfong Mayxay, Chamsai Pientong, Sirinart Aromseree, Petchaboon Poolphol, Tiengkham Pongvongsa, Nanthasane Vannavong, Hans J. Overgaard

**Affiliations:** 1Department of Environmental Engineering and Management, Asian Institute of Technology; Pathumthani 12120, Thailand; oshipin@ait.asia; 2Unité de la Génétique Fonctionnelle des Maladies Infectieuses, Institut Pasteur, CNRS UMR 2000, 75015 Paris, France; rpaul@pasteur.fr; 3Department of Public Health and Clinical Medicine, Umeå University, 90187 Umeå, Sweden; joacim.rocklov@umu.se; 4Department of Biostatistics and Epidemiology, University of North Texas Health Science Center, North Texas, Fort Worth, TX 76107, USA; Mdubydul.Haque@unthsc.edu; 5Department of Microbiology, Faculty of Medicine, Khon Kaen University, Khon Kaen 40002, Thailand; siddikur@brur.ac.bd (M.S.R.); chapie@kku.ac.th (C.P.); sirinar@kku.ac.th (S.A.); hans.overgaard@nmbu.no (H.J.O.); 6Department of Statistics, Begum Rokeya University, Rangpur 5402, Bangladesh; 7Institute of Research and Education Development (IRED), University of Health Sciences, Ministry of Health, Vientiane 43130, Laos; mayfong@tropmedres.ac; 8Lao-Oxford-Mahosot Hospital-Welcome Trust Research Unit (LOMWRU), Microbiology Laboratory, Mahosot Hospital, Vientiane 43130, Laos; 9Centre for Tropical Medicine and Global Health, Nuffield Department of Clinical Medicine, Old Road Campus, University of Oxford, Oxford OX3 7LG, UK; 10The Office of Disease Prevention and Control Region 10(th), Ubon Ratchathani 34000, Thailand; siapoolphol@gmail.com; 11Savannakhet Provincial Health Department, Savannakhet 13000, Laos; tiengkhampvs@gmail.com; 12Champasak Provincial Health Office, Pakse 1600, Laos; anandafet@gmail.com; 13Faculty of Science and Technology, Norwegian University of Life Sciences, P.O. Box 5003, 1430 Ås, Norway

**Keywords:** exposure, epidemiology, health status indicators, spatial analysis, susceptibility

## Abstract

Dengue is a continuous health burden in Laos and Thailand. We assessed and mapped dengue vulnerability in selected provinces of Laos and Thailand using multi-criteria decision approaches. An ecohealth framework was used to develop dengue vulnerability indices (DVIs) that explain links between population, social and physical environments, and health to identify exposure, susceptibility, and adaptive capacity indicators. Three DVIs were constructed using two objective approaches, Shannon’s Entropy (SE) and the Water-Associated Disease Index (WADI), and one subjective approach, the Best-Worst Method (BWM). Each DVI was validated by correlating the index score with dengue incidence for each spatial unit (district and subdistrict) over time. A Pearson’s correlation coefficient (r) larger than 0.5 and a *p*-value less than 0.05 implied a good spatial and temporal performance. Spatially, DVI_WADI_ was significantly correlated on average in 19% (4–40%) of districts in Laos (mean *r* = 0.5) and 27% (15–53%) of subdistricts in Thailand (mean *r* = 0.85). The DVI_SE_ was validated in 22% (12–40%) of districts in Laos and in 13% (3–38%) of subdistricts in Thailand. The DVI_BWM_ was only developed for Laos because of lack of data in Thailand and was significantly associated with dengue incidence on average in 14% (0–28%) of Lao districts. The DVI_WADI_ indicated high vulnerability in urban centers and in areas with plantations and forests. In 2019, high DVI_WADI_ values were observed in sparsely populated areas due to elevated exposure, possibly from changes in climate and land cover, including urbanization, plantations, and dam construction. Of the three indices, DVI_WADI_ was the most suitable vulnerability index for the study area. The DVI_WADI_ can also be applied to other water-associated diseases, such as Zika and chikungunya, to highlight priority areas for further investigation and as a tool for prevention and interventions.

## 1. Introduction

Dengue fever is a rapidly spreading arboviral disease globally and is of public health concern in many tropical and subtropical countries, and in some temperate countries. The disease is caused by the dengue virus (DENV), which belongs to the genus Flavivirus transmitted to humans principally by *Aedes aegypti* and *Aedes albopictus* mosquitoes through blood feeding. The number of dengue cases reported to the World Health Organization (WHO) has increased eightfold over the past two decades, from 505,430 cases in 2000 to over 5.2 million in 2019 (WHO, 2021) and modelling estimates suggest there may be as many as 96 million apparent and 294 million inapparent dengue infections [1,2]. The risk of exposure to vector-borne disease is spatiotemporally heterogeneous due to the variability of climate, mosquito densities, and the physical environment [3,4]. Dengue outbreaks in South East Asian countries are exacerbated by climate change and modification in landcover because of urbanization, deforestation, and agricultural intensification [5,6]. The dengue burden has been disproportionately affecting socioeconomically disadvantaged populations, who often have less capacity to invest in resilience-building and adaptation activities [7,8,9].

To lessen the burden of disease, it is crucial to identify populations who are vulnerable to the high burden of dengue, whether it is because of their socio-environmental living conditions or because of poor health systems. Vulnerable populations can be defined as those who are economically underprivileged, have low income, are elderly or children, have chronic health conditions, and who face significant disparities in healthcare [10]. A vulnerability assessment is an approach used to describe the potential for harm from existing hazards in susceptible populations, and their adaptive capacity on a local to an international scale to help relevant decision-making processes [11]. Dengue risk has predominantly been estimated and mapped both retrospectively and prospectively at global [12,13], country or provincial levels [14]. However, vulnerability studies at smaller administrative levels such as the district/subdistrict level are scarce in South East Asia. Fine scale vulnerability mapping can help communities to act and create awareness about vector control and dengue prevention.

The vulnerability of populations to infectious disease has been mapped at the global level by the Infectious Disease Vulnerability Index (IDVI) [13] and the Water-Associated Disease Index (WADI) [12,14]. The WADI was first used for dengue vulnerability in Malaysia using freely available published data such as living conditions, population characteristics, climate, land use, and landcover. The WADI index has also been effectively used to map dengue vulnerability in Vietnam [15], Brazil [16], and Jamaica [17]. Several other dengue vulnerability assessment studies have been conducted in Bhutan [18], India [19], Malaysia [20], and Thailand [21], but using few criteria and excluding human habitation conditions including housing, water and hygiene, and literacy rates. 

A vulnerability assessment approach for dengue fever is needed that can also be used to map vulnerabilities at different spatial scales. The objectives of this study were to develop and compare dengue vulnerability indices using objective and subjective approaches and to select the best index to map dengue vulnerability in four provinces of Laos and Thailand.

## 2. Materials and Methods

### 2.1. Study Area

The selected study areas were Savannakhet and Champassak provinces in Lao PDR (Laos) and Mukdahan and Ubon Ratchathani provinces in northeastern Thailand (Figure 1). There were 25 administrative units in the Lao provinces (10 districts in Champasak and 15 in Savannakhet) and 272 in Thailand (53 subdistricts in Mukdahan and 219 in Ubon Ratchathani). These four provinces are quite similar in terms of culture, language, and history; however, they differ in socioeconomic and political conditions and are expected to vary in climate vulnerability, adaptive capacity, geographical and ecological diversity, and socioeconomic status. The region has a tropical climate with a dry cool season from mid-October to mid-February, a dry hot season from mid-February to mid-May and a monsoon rainy season from mid-May to mid-October with higher rainfall, humidity, and temperatures.

### 2.2. Conceptual Framework

We used the three components that vulnerability consists of, i.e., exposure, susceptibility, and adaptive capacity, including indicators of each component that were specifically selected for this study (Figure 2). In this framework, exposure illustrates conducive conditions for the survival of the vector and transmission of DENV in the environment. Individual susceptibility explains the physical sensitivity of an individual, including factors such as age and gender. Community susceptibility includes factors of housing quality/conditions, water, and sanitation. Adaptive capacity includes the conditions that impact the resilience of populations, a concept described here as the capacity to prevent, respond to, and cope with dengue exposure [22].

Two new dengue vulnerability indices were developed by applying different weighting systems (for the indicators) based on an objective approach (Shannon’s Entropy, SE, explained in Section 2.5.3) and a subjective approach (Best-Worst Method, BWM explained in Section 2.5.2). These approaches were then compared with an existing dengue vulnerability index (WADI, explained in Section 2.5.1) that uses an objective equal-weighting system. To assess the effectiveness of the developed vulnerability indices (SE and BWM) to identify the areas at risk and the existing WADI index, the Pearson correlation (*p*-value < 0.05) between the indices and dengue incidence was assessed. 

### 2.3. Data Collection

Historical daily reported dengue case data were acquired from 2003 to 2019 from provincial health departments in the study provinces. For Thailand, subdistrict level dengue cases were reported as dengue fever (DF), dengue hemorrhagic fever (DHF), and dengue shock syndrome (DSS). In Laos, dengue cases at the district-level were reported as DF, DHF, and DSS up to and including 2009 and thereafter followed the new classification of dengue with or without warning signs and severe dengue [7]. District and subdistrict level population data from 2002 to 2019 for Laos and Thailand were acquired from official web portals of the national departments of statistics [23,24]. Monthly dengue incidence per 100,000 persons was calculated for each spatial unit by dividing the number of cases by its total population and multiplying it by 100,000. Socioeconomic data for Laos including toilet type, living conditions or housing quality, mean distance to hospital (km), literacy rate (%), and poverty incidence (%) were acquired from census data available at official web portals of the national departments of statistics [23]. A detailed description of collected datasets for constructing dengue vulnerability indices is presented in Table 1.

Monthly cumulative precipitation and mean temperature data were obtained from the fifth-generation European Centre for Medium-Range Weather Forecasts atmospheric reanalysis (ERA5) of the global climate and aggregated at the district/subdistrict level [42]. Land use and land cover data were obtained from Landsat ETM+ for 2002–2003, Landsat TM for 2004–2011, and Optical Land Imager (OLI) and from Thermal Infrared (TIR) for 2013–2015 [30] (Figure 3) (Appendix A: Land use and landcover change in Champasak and Savannakhet provinces in Laos and Mukdahan and Ubon Ratchathani provinces in Thailand between 2002 and 2019).

### 2.4. Determinants and Indicators of Vulnerability Index

Due to the multidimensionality of determinants, indicators are commonly used as proxies to simplify and integrate the diverse measures into a composite index (Table 2). To overcome this multidimensionality, the indicators used in BWM and WADI were categorized and given a score between 0 and 1, representing a range from low to high exposure, susceptibility, and adaptive capacity (Table 2). For SE, indicators were divided into sub classes according to their importance for dengue vector and virus development and transmission based on current literature. Weights for each indicator were calculated based on the number of cases occurring in each subclass out of the total population (Appendix A). For SE, the population of each district was categorized into classes according to the values of indicators, e.g., for temperature the population was divided into two subclasses, 1 = population living in districts with suitable temperature range of 24–29 °C and 0 = population living in districts with a lower suitable temperature range <24 °C and >29 °C [43] (Appendix A). 

#### 2.4.1. Exposure

We considered climate, land use/land cover, and the human environment and their change as the major exposure indicators from 2003 to 2019 (Table 2). Climatic variables including temperature and rainfall are the major drivers of dengue infection and directly affect the vector life cycle, feeding activity, biting rates, and virus incubation period [44,45,46,47]. Precipitation provides outdoor oviposition sites for vectors and humid conditions favor mosquito survival. By contrast, heavy rainfall can flush breeding sites of mosquito immatures [48].

Mean temperatures between 24 °C and 29 °C and total precipitation between 0 mm and 300 mm were given a value of 1 and values outside of these specified ranges were categorized as 0. A one-month lag time was used in the analysis due to the delayed effect of climatic parameters found in a retrospective study of dengue and environment that we carried out in the same provinces [43].

Land use and landcover modified by humans, such as built-up areas, contribute to exposure by providing indoor oviposition sites in artificial containers for the primary vector *Aedes aegypti*. Trees and other plantations provide oviposition sites for the secondary vector *Aedes albopictus*. All land use and landcover determinants were ranked on the continuous scale from 0 to 1: built-up areas were considered highly vulnerable and rated as “1”, agricultural lands/paddy fields were rated as “0.25”, rubber plantation as “0.5”, and deforested area as “0.25”, while forest and wetlands were considered not vulnerable to dengue and rated as “0” (Table 2) [14]. Densely populated human environments create ideal conditions for dengue outbreaks [49,50]. The population density was calculated for each district in selected provinces based on population census data [23,24], and classified into zero, low (0–200 people per km^2^), moderate (200–400 people per km^2^), and high density (>400 people per km^2^). The geographical distribution of indicators of exposure are presented as maps in Figure 2 and Appendix A.

#### 2.4.2. Susceptibility

In this study, we considered individual and collective factors of population susceptibility to dengue and their change from 2003 to 2019 (Table 2). Socioeconomic and demographic factors have been reported to affect dengue transmission [51]. Physiological parameters such as the individual’s immunity and previous exposure are also important, but such data are usually not published or easily accessible. Therefore, age was used as a proxy for immunity and previous exposure. Age is related to serious forms of dengue; children less than 15 years and adolescents are relatively more susceptible to dengue hemorrhagic fever and dengue shock syndrome (DHF/DSS), independently of other factors [52]. Historically, dengue cases occurred in the age groups less than 15 years and over 60 years, as is shown in the Appendix A. Dengue cases increase in the age group >60, suggestive of decreasing immune competence and higher frailty. 

Living conditions such as poor housing quality, defined as the lack of window and door screens, commonly found in underprivileged areas, allows free passage for mosquitoes between the indoor and outdoor areas [2]. Houses built with bamboo and wood, porous floors, unplastered walls, and bathrooms without tiles can cause increased indoor humidity, conducive to vector survival [35]. We calculated percentage housing categories for each district—the districts were assigned with the most abundant housing category value [35]. Districts with most houses constructed with concrete and wood were given the lowest score of 0.25, a score for houses with wood 0.5, and houses with wood and bamboo were assigned the highest score of 1. Dengue susceptibility is increased in urban areas, particularly in slums with inadequate waste disposal and toilets, and in peri-urban areas with slower economic development and poor housing quality [9,53,54]. The geographical distribution of susceptibility indicators in 2003 and 2019 are presented as maps in the Appendix A.

#### 2.4.3. Adaptive Capacity

We considered women’s literacy rates, proximity to health care centers, and poverty and their change from 2003 to 2019 as collective factors of the population’s adaptive capacity towards dengue. The adaptive capacity indicators reflect the ability of populations to cope with or prevent dengue outbreaks [22]. Women play important roles in households. They commonly manage house conditions, waste, water, the family’s health care management, etc. Households with low female literacy rate were associated with increased risk of *Aedes* oviposition sites around households [55]. Good access to healthcare can reduce the susceptibility especially to complications, and increase early diagnosis of disease and immediate medical care to reduce morbidity and thus can reduce the susceptibility of the population; therefore, distance to a hospital was included as an indicator of adaptive capacity [56]. The geographical distribution of adaptive capacity indicators in 2003 and 2019 are presented as maps in Appendix A.

Normalization of determinants was performed by standardizing the data to a value from 0 to 1 based on an approach used for the human development index (HDI). The HDI was developed to measure key dimensions of human development, including health, education, and standard of living [57]. 

The contribution in percent of each of the three determinants—exposure, susceptibility, and adaptive capacity—to the total vulnerability score (DVI_WADI_) in each country was calculated by dividing their final average scores with the respective DVI average for Laos and Thailand.

### 2.5. Index Construction

An index was constructed using estimated weights for each indicator using the three different approaches, i.e., WADI, BWM, and SE. The index was constructed using linearly weighted averages (LWA) to combine the indicators of the determinants Equation (1) [58]. The estimated weights of each index were used along with standardized criteria as input for the LWA. The total score was obtained as the product sum of each indicator and its weight as follows:(1)$${\mathrm{DVI}}_{\mathrm{indexname}}=\overset{n}{\underset{i=1}{\sum }}w_{i}x_{i}$$
where DVI is the Dengue Vulnerability Index, $w_{i}$ is the weight of factor *i*, and $x_{i}$ is the criterion score [59]. Equation (1) is the common expression for all three indices, where the index name represents either WADI, BWM, or SE. 

#### 2.5.1. Index Based on Water Associated Disease Index (DVI_WADI_)

We used the conceptual framework of WADI developed by Dickin et al. (2013). In the WADI framework, the vulnerability index was developed based on the exposure, susceptibility, and adaptive capacity indicators with weights of 3, 1, and 1, respectively [14]. All the indicators within exposure, susceptibility, and adaptive capacity were assumed to have an equal weight and aggregated to form a composite index using an arithmetic average.

#### 2.5.2. Index Based on Best-Worst Method—BWM (DVI_BWM_)

Subjective approaches to map vulnerability have been criticized for two reasons. First, they can be subjectively biased by the decision makers’ opinions, and second, they compare indicators among different domains such as exposure and sensitivity, i.e., rubber plantations versus female literacy rates. Similarly, objective-based approaches are criticized for not including expert knowledge in developing such indices, even though expert opinions can also be biased. The BWM is an advanced form of the most frequently used subjective approach, the analytical hierarchy process, which allows comparisons in specific domains by reducing the comparisons of criteria and limiting subjectivity bias [60].

This method is used to solve different real-world problems and is comparable to the Analytical Hierarchy Process that uses several evaluation criteria [60]. Weights calculated using BWM are based only on the reference comparisons (comparing the best and the worst indicators to the others). To calculate the weights using BWM, the following steps are required.
i.Define a set of decision indicators that will be used to derive a decision process, $\left\{C_{1},\;C_{2}\ldots ,\;\;C_{n}\right\},$ where C indicates the multiple indicators of each determinant (exposure, susceptibility, and adaptive capacity) (Table 2).ii.Define the best (most important) and the worst (least important) indictor for each determinant.iii.Experts determine the preference of the best and worst indicator over all the other indicators using a number between 1 to 9 (1 = Worst and 9 = Best). The resultant vector of Best-to-Others would be
$$A_{B}=\left(a_{B1},\;a_{B2},\ldots ..a_{Bn}\right)$$
where $a_{Bj}$ indicates the preference of the best criterion B over criterion *j*, and $a_{BB}$ = 1.iv.Preference of all the criteria over the worst criterion was determined using a value between 1 and 9. The resultant vector for Others-to-Worst would be
$$A_{W}={\left(a_{1W},\;a_{2W},\ldots ,\;\;a_{nW}\right)}^{T},$$
where $a_{jW}$ indicates the preference of the criterion j over the worst criterion W and $a_{WW}$ =1.v.Estimate the optimal weights $w_{1}^{*},\;w_{2}^{*},\ldots ,\;\;w_{n}^{*}.\;$
The optimal weight for the indicator is the one where, for each pair of $\frac{w_{B}}{w_{j}}\;\mathrm{and}$ $\frac{w_{j}}{w_{w}},$ we have $\frac{w_{B}}{w_{j}}=a_{Bj}$ and $\frac{w_{B}}{w_{w}}=a_{jW}$. To find an optimal solution, the maximum absolute differences $\left|\;\frac{w_{b}}{w_{j}}-a_{Bj}\right|$ and $\left|\frac{w_{j}}{w_{W}}-a_{jW}\right|$ for all *j* is minimized. Based on the non-negativity characteristic and sum condition of the weights, the following problem was also formulated:
$$\min\underset{j}{\max}\;\left\{\left|\frac{w_{b}}{w_{j}}-a_{Bj}\right|,\left|\frac{w_{j}}{w_{W}}-a_{jW}\right|\;\right\}$$
s.t.
(2)$$\underset{j}{\sum }wj=1$$$w_{j}\geq 0$, for all j.Hence, the problem in Equation (2) can be transferred to the linear problem:(3)$$\begin{array}{l} \left|\frac{w_{b}}{w_{j}}-a_{Bj}\right|\;\leq \xi ,\;\mathrm{for}\;\mathrm{all}\;j\\ \left|\frac{w_{j}}{w_{W}}-a_{jW}\right|\leq \xi ,\;\mathrm{for}\;\mathrm{all}\;j\\ \underset{j}{\sum }wj=1\\ w_{j}\geq 0,\;\mathrm{for}\;\mathrm{all}\;j \end{array}$$

Solving Equation (3), the optimal weights $(w_{1}^{*},\;w_{2}^{*},\ldots ..,\;w_{n}^{*}$) and $\xi^{*}$ are obtained [60].

The pairwise comparisons that we carried out in this research using BWM methods were integrated with linear weighted average to construct the index. The contributing weight for each determinant of vulnerability (exposure, susceptibility, and adaptive capacity) was also calculated (Table 3). 

The BWM method was not used for Thailand due to the unavailability of the data for comparisons to calculate indicator weights.

#### 2.5.3. Index Based on Shannon’s Entropy—SE (DVI_SE_)

Shannon’s Entropy has recently been used to map the susceptibility of dengue in Chicago, IL, USA [61]. SE relies on objective base information that can be used to determine the disorder degree of the information in the decision space of the variables for a particular decision problem and can minimize the subjective bias of decision maker opinion. Shannon’s Entropy is a measure of the amount of information held in data using probability theory. It indicates that a broader distribution contains more uncertainty than a sharply peaked one does. In other words, the most important factor is the discriminating density $d_{ij}$ of the subclasses among all indicators (Appendix A). The determination of weight for each indicator was objectively assessed because it was decided by the distribution of dengue cases and not by personal opinions, as in the BWM method. According to entropy theory, the dengue case densities ($d_{ij}$) in each class were calculated by Equation (4).
(4)$$d_{ij}=\frac{DP_{ij}}{TP_{ij}}\;i=(1,\;2,\;\ldots .,\;m),\;j=1,\;2,\;\ldots .,\;n)$$
where $DP_{ij}$ is the number of dengue cases in jth class of indicator i and $TP_{ij}$ is the average population in subclass *ij*.

The densities of the subclasses ($d_{ij}$) were normalized to yield a non-negative index, denoted by $p_{ij}$, Equation (5).
(5)$$p_{ij}=\frac{d_{ij}}{\sum_{j}^{n}\;d_{ij}}\;$$

The entropy value $H_{j}$ is given by Equation (6).
(6)$$H_{j}=\;-e\;\sum_{j=1}^{m}p_{ij}ln(p_{ij})\;i\;=\;(1,\;2,\;\ldots .,\;m)$$
where e = 1/*ln*(*n*) is a constant that guarantees 0 ≤ $H_{j}$ ≤1. 

The objective weight ($w_{ij}$) of each factor is given by Equation (7).
(7)$$w_{ij}=\;\frac{1-H_{j}}{m-\sum_{j=1}^{m}H_{j}}\;(i\;=\;1,\;2,\;\ldots .,\;m;\;j\;=\;1,\;2,\;\ldots .,\;n;)$$

The value of $1-H_{j}$ is known as the degree of diversification $d_{ij}$ of the jth index, which describes the divergence degree of the inherent information of each indicator of the determinant. The larger the value of $d_{ij}$, the higher the variation in the $j_{th}$ index. 

According to entropy theory, if the dengue case densities ($d_{ij}$) of the subclasses of an indicator are the same, the indicator can be excluded from the causative system because the frequency of historical dengue cases does not change in various subclasses. The DVI_WADI_ was used to create aggregate maps of vulnerability at five-month intervals (April and September) and the change in vulnerability was also mapped for similar months for the years 2003 and 2019.

### 2.6. Validation

Validation of a vulnerability index for dengue is a challenging task because vulnerability is a composite of exposure, susceptibility, and adaptive capacity that can prevail without virus transmission. However, monthly dengue incidence per 100,000 persons was used as a proxy measure of vulnerability in this analysis. The three of the developed indices were validated, by evaluating the association between dengue incidence and vulnerability values, and aggregated at the district level. Pearson’s correlation coefficients and *p*-values were used to identify the significant association. 

## 3. Results

### 3.1. Dengue Incidence

There were 48,852 cases of dengue recorded during the study period with an annual average of 2874 cases and an annual incidence of 12.7 cases per 100,000 persons in selected provinces of Laos. Based on an outbreak definition of >300 cases per 100,000 persons [62], there were four major outbreaks in Laos, namely in 2003, 2010, 2013, and 2016, with greater intensity as compared to other years. The two provinces in Thailand recorded relatively fewer cases than the Lao provinces with a total of 39,444 cases with an annual average of 2320 cases (Table 4). Using the same outbreak definition as above, 2003, 2010, 2013, 2015–2016, and 2018–2019 were classified as the outbreak years in Thailand. 

### 3.2. Associations between Vulnerability Indices and Dengue Incidence

The association between the calculated indices and monthly dengue incidence per 100,000 persons per administrative area (district in Laos and subdistrict in Thailand) varied substantially from year to year and among indices. In Laos, all indices were generally poor with mean correlations varying between 0.23–0.5 (Figure 4). In Thailand, DVI_WADI_ was an exception where the mean significant correlation was 0.85 (Figure 4 and Appendix A). For Laos, susceptibility indicators (Table 2) were used at the community and individual level, but in Thailand, the DVIs were developed only with individual susceptibility because of limited available data. SE performed well with comprehensive data in Laos, but WADI produced relatively good results with limited data in Thailand. 

#### 3.2.1. Laos

In Laos, on average, annually 19% (6 out of 25) and 22% (6 out of 25) of the districts had significant correlations (*p*-value < 0.05) when applying the DVI_WADI_ and DVI_SE_, respectively (Appendix A) and respective percentages out of the 25 districts of Savannakhet and Champasak in Laos and the 272 sub-districts of Mukdahan and Ubon Ratchathani in Thailand during 2003–2019. These two indices were also good in associating dengue incidence during outbreak years in Laos (2003, 2010, 2013, and 2019) as assessed by the percentage of districts with a significant association, which were 21% for DVI_SE_ and 14% for DVI_WADI_ (Appendix A). The DVI_SE_ was significantly associated with dengue incidence during all the outbreak years except in 2010 (Figure 4a, and Appendix A). In contrast, DVI_BWM_ and DVI_WADI_ showed significant association in 2010 (outbreak year) and years other than outbreak (Appendix A).

#### 3.2.2. Thailand

In Thailand, on average 27% (73 out of 272) and 13% (35 out of 272) of the subdistricts had significant correlations (*p*-value < 0.05) when applying the DVI_WADI_ and DVI_SE_, respectively (Appendix A). Only DVI_WADI_ was found to be significantly associated with dengue incidence during outbreak years in Thailand (2003, 2010, 2013, 2015, 2016, 2018, and 2019) (Appendix A). For Thailand, DVI_WADI_ was developed with only individual susceptibility indicators due to lack of data at the community level (living conditions and water and hygiene) and with two indicators of adaptive capacity (mean distance to hospital and poverty). However, even when the DVI_WADI_ was developed with limited data, it showed a significant association with dengue incidence in all years (Figure 4 and Appendix A). 

### 3.3. Spatiotemporal Variation of Dengue Vulnerability

#### 3.3.1. Spatial Differences 

Dengue vulnerability varied temporally and spatially over the study period (Figure 5, Figure 6 and Figure 7). Generally, the highest vulnerability clusters were observed in administrative units with a high proportion of built-up areas and those undergoing new developments, such as establishing new settlements, improved road networks, and dam constructions in forests with associated resettlement of workers and inhabitants in remote areas. There was moderate to low vulnerability observed in sparsely populated areas mostly covered with forest and crops (compare Figure 2 and Figure 5b). 

In Laos, the highest vulnerabilities were observed in the provincial capitals of Savannakhet and Pakse. The western district of Champasak and southern districts of Savannakhet showed significant changes in vulnerability due to the recent development activities in these areas reducing forest cover (Figure 2 and Figure 7). In Thailand, the highest vulnerabilities were also observed in the provincial capitals of Mukdahan and Ubon Ratchathani. Central parts of Mukdahan that underwent extensive deforestation and creation of rubber plantations and a shift in agriculture from crops to rubber or other tree plantations also showed high dengue vulnerability. The impact of land cover change was not visible in the overall vulnerability of Ubon Ratchathani, but there was an increase in exposure over the study period due to changes in temperature and rainfall indicators (Appendix A).

#### 3.3.2. Temporal Differences

Dengue vulnerability varied seasonally, similarly to dengue incidence, with higher vulnerability in the rainy season in all provinces (Figure 6). In Lao provinces, mean vulnerabilities remained high throughout the year compared to the Thai provinces, with maximum vulnerability values during July in Champasak and August in Savannakhet. In both provinces of Thailand, the vulnerability started to increase from January towards the highest vulnerability during June. Maximum vulnerability and dengue incidence were observed from May to July (Figure 6). In Ubon Ratchathani, the lowest vulnerability values were noted in the post-monsoon months from October to December. In Mukdahan province, relatively higher vulnerability was also recorded in November.

Over the study period, average dengue vulnerability in Lao provinces remained higher than Thai provinces (Figure 7a). Compared to 2003, dengue vulnerability in 2019 decreased in the northern, north-eastern, and central districts of Savannakhet and northern and north-eastern districts of Champasak. In Savannakhet, dengue vulnerability increased in western districts and the western district of Champasak in 2019 compared to 2003. Overall vulnerability in the Lao provinces generally decreased except for an increase in western districts of both provinces. In the Thai provinces, during 2003, high dengue vulnerability values were observed in north-western Mukdahan and central and eastern Ubon Ratchathani. In 2019, average vulnerability decreased in both provinces. In Ubon Ratchathani, average vulnerability remained high in eastern districts bordering Champasak. The overall average vulnerability decreased in both provinces. 

In Laos, dry season dengue vulnerability (represented by the month of April) (Figure 7b) decreased between 2003 and 2019 in eastern districts and increased in western and southern districts of Savannakhet. The rainy season vulnerability (represented by the month of September) decreased during the same period in northern Savannakhet, but significantly increased in southern districts (Figure 7c). In Champasak province, both dry and rainy season vulnerabilities remained unchanged between 2003 and 2019, except for increases in one eastern district (Figure 7b,c). In Thailand, dry season dengue vulnerability decreased between 2003 and 2019 in northern Mukdahan and Ubon Ratchathani provinces (Figure 7b). However, the rainy season dengue vulnerability increased in Mukdahan province except for a decrease in one northern subdistrict. In Ubon Ratchathani, the largest decreases in rainy season vulnerability were observed in eastern and southern subdistricts over the study period (Figure 7c). 

#### 3.3.3. Change in DVI_WADI_ Components

Over the study period, exposure to dengue increased in some western districts of both provinces in Laos, while in Thailand, dengue exposure remained unchanged in Ubon Ratchathani and increased in northern Mukdahan (Appendix A). Susceptibility of the population towards dengue also increased in most districts/subdistricts in both countries (Appendix A). Adaptive capacity decreased in northern and north-western districts of Savannakhet and in Champasak. Adaptive capacity remained unchanged in southern districts of Savannakhet except for a decrease in one. In Ubon Ratchathani, adaptive capacity decreased in most subdistricts and increased only in a few subdistricts. The adaptive capacity in northern Mukdahan increased but decreased in southern Mukdahan (Appendix A).

In Laos, the exposure component score contributed on average 11%, the susceptibility score 47%, and adaptive capacity 42% to the full vulnerability score. In Thailand, the exposure component score contributed 56%, the adaptive capacity 27%, and susceptibility 17% to the vulnerability score.

## 4. Discussion

We developed and compared three dengue vulnerability indices using objective and subjective weighting approaches and selected the best index to map dengue vulnerability in border areas between Laos and Thailand. The three vulnerability indices were DVI_WADI_, DVI_SE_, and DVI_BWM_. Of the three indices assessed here, the DVI_WADI_ performed best with a significant correlation with dengue incidence (Pearson correlation coefficient, r > 0.5 and *p* < 0.05) in areas with high dengue incidence during outbreak and non-outbreak years. The DVI_SE_ performed better than DVI_BWM_, but not in the highly vulnerable provincial capitals of Laos (Figure 5). The DVI_BWM_ overestimated vulnerability in administrative units that experienced very low dengue incidence and underestimated in highly vulnerable provincial capitals with high dengue incidences (Figure 5). However, DVI includes conditions of exposure, susceptibility, and adaptive capacity that can also occur without DENV transmission [50,63].

The DVI_WADI_ was a better index compared to DVI_SE_ and DVI_BWM_ to map vulnerabilities because it presented higher spatio-temporally significant correlations with dengue incidences (*r*> 0.5 and *p* < 0.05) in the study areas in both countries. The validation of DVI_WADI_ results showed better performance in Thailand than in Laos with a significant positive correlation between vulnerability and dengue incidence in 16 out of 17 years in 27% (15–53%) of the subdistricts on average in non-outbreak years and 36% (22–53%) in outbreak years. 

### 4.1. DVI_WADI_ Spatial Variations

The highest annual average DVI_WADI_ was observed in districts with major urban centers and high population densities (Figure 5). In the Thai provinces, clusters of highly vulnerable districts were in and around the provincial capitals, and in districts that mostly consisted of forest and tree plantations of cashew and rubber. In Laos, the provincial capitals were highly vulnerable, and the rest of the districts were moderately to highly vulnerable. The temporal variations during 2003–2019 in vulnerability (Figure 7) also highlighted the impact of land covers and their change.

### 4.2. Change in DVI_WADI_ during 2003–2019 

The combination of socioeconomic and environmental indicators affect dengue vulnerability [64,65]. Environmental indicators of exposure including climate and land cover play a role at a larger scale, while socioeconomic indicators of susceptibility and adaptive capacity define the risk of communities toward dengue infection. 

In the selected provinces, vulnerability decreased in 67% of districts in Laos and in 87% of the subdistricts in Thailand between 2003 and 2019 (Figure 7). This likely reflects the general improvement in indicators of susceptibility, specifically hygiene facilities, and an increase in adaptive capacity with a decrease in poverty incidence because of better socioeconomic status and increased female literacy rates (Appendix A). A study from southern Mexico [55] also indicated that the households with poor socioeconomic conditions and low educated mothers tend to have high risk for dengue with more larval breeding containers.

Dengue vulnerability increased in the remaining areas, i.e., in 33% of the Lao districts and 13% of the Thai subdistricts (Figure 7). This increase in dengue vulnerabilities was mainly linked with the increase in exposure in the eastern districts of Savannakhet and Champasak and in the whole of Mukdahan. 

In Laos, the significant land cover modification included forest clearing, forest and agriculture land conversion to rubber plantations, dams, settlements, and road construction (Figure 2 and Figure 7) [66]. In 2003, the southern district of Savannakhet province consisted of forest and experienced low vulnerability in the dry season and high vulnerability in the monsoon season. In 2019, these forests were cleared mainly for dams construction and other land-uses [67], which ultimately increased the human population density, causing an increase in vulnerability in this district. The increase in vulnerability in the southern districts of Savannakhet was pronounced when comparing the rainy season in 2003 to that of 2019 (Figure 7c). Similarly in Malaysia [68], deforestation was found to be associated with increased dengue cases. 

In Thailand, Mukdahan province was the least affected by dengue, although the rainy season vulnerability increased between 2003 and 2019 while remaining unchanged in the dry season. Changes in dengue vulnerability in Mukdahan were associated with climatic changes and extreme climatic events (Appendix A), especially flooding in the rainy season from 2011 onwards (Appendix A). Land cover in Mukdahan also changed, for example, with increased deforestation and conversion to agriculture and rubber plantations (Figure 2) [69]. 

### 4.3. DVI_WADI_ Temporal Variations

DVI_WADI_ presented strong seasonal variations in dengue vulnerability, driven by the climate variation in both countries across the year. In both countries, high vulnerability was related to the rainy season that spans from May to October; this seasonal cycle differs among selected provinces (Figure 6). This seasonality of dengue is widely reported in South East Asian countries [12,70,71]. 

### 4.4. Change in Determinants of DVI_WADI_


The three determinants of dengue vulnerability independently changed over the study period.
i.Exposure: All four indicators of exposure including land cover, temperature, rainfall, and population density changed over the study period. An increase in exposure/risk with landcover and climate is also reported in other South East Asian countries such as Malaysia [68], Vietnam [72], Indonesia [73], and Timor-Leste [74]. The increasing risk with climate is also well reported for different parts of Thailand [21,51,75,76,77,78]. On average, exposure contributed 56% to the DVI_WADI_ score in Thailand and only 11% in Laos. The relatively higher contribution of exposure to dengue vulnerability in Thailand could be a reason for the overall high correlation between DV_IWADI_ and dengue incidence (R^2^ = 0.73 and *r* = 0.85), even if data for susceptibility and adaptive capacity determinants were limited (Figure 4). On average, exposure contributed 56% to the DVI_WADI_ score in Thailand and 11% in Laos. Both countries went through similar environmental changes, but there were marked differences in population density; for example, population densities remained low in Savannakhet (below 100 person/km^2^ between 2003–2019). The low exposure in Laos can be an artifact because of larger spatial units considered as compared to Thailand where subdistricts are much smaller than districts in Laos.ii.Susceptibility: People’s susceptibility to dengue decreased in both countries with improved water and hygiene facilities (Appendix A). In Laos, the population with improved access to water and sanitation increased by 18% between 2005 and 2010 alone and from 45% to 63% nationwide, exceeding the Millennium Development Goals target of 54% [79]. Similarly, in Thailand, nearly 93% of the population has access to improved hygiene and 96% to drinking water [80]. In Laos, susceptibility on average contributed 47% and in Thailand, only 17% to the DVI_WADI_ score, which underlines the different living conditions and vulnerable population age groups in the study sites of these two countries. The relatively higher contribution of susceptibility and limited data (censuses in 2005 and 2015) in Laos might be the reason for an overall lower correlation between DVI_WADI_ and dengue incidence (R^2^ = 0.22 and r = 0.5) (Figure 4).iii.Adaptive capacity: In Laos, the adaptive capacity on average contributed 42% to total DVI_WADI_ score and in Thailand only 27%. Poverty and female literacy rates were the most critical indicators of adaptive capacity that can effectively help to reduce the dengue burden. Research from southern Brazil [81] reported 23–32% reduction in dengue cases with an increase in mean income from approximately USD 100–200 to USD 200–300. Poverty incidence rates decreased in Laos provinces between 2005 and 2015 (Appendix A). In Champasak, poverty incidence dropped to 21% from 28% and in Savannakhet from 49.3% to 32.2%. In Thailand, the poverty incidence rate also decreased between 2003 and 2019, from 8% to 1.3% in Mukdahan and from 2.5% to 1.1% in Ubon Ratchathani [24]. The average female literacy rate between 2005–2015 increased from 44.6% to 48.7% in Champasak province and from 38.8% to 42.6% in Savannakhet province [23]. The average female literacy rates in Mukdahan and Ubon Ratchathani in 2000 were 86% and 90%, respectively, and increased to 95% in both provinces in 2019 [24]. The average distance to a hospital or health care facility is a crucial indicator of community adaptive capacity. However, this distance remained unchanged in Laos between the 2005 and 2015 censuses [23]. In Laos, except for urban centers, the minimum average distance to the nearest health facility was 10–20 kilometers. The lack of geographical coverage of the health system was due to a sparse population in the country, with approximately 80% of the population living in rural areas and engaged in agriculture [82]. The current health facility network in Thailand showed good coverage, with the nearest health care facility between 1 and 10 kilometers.

Vulnerability indices simplify the complex real-world information and provide a science–policy interface for decision makers. The DVI maps developed in this study (Figure 7) identified areas needing vector control and communities with deprived health services. Such maps could inform the health department on areas in most need of prevention and intervention activities. Along with previous studies [12,15,16], this study emphasizes the role of the WADI as a holistic tool to highlight key factors and linkages that play a role in DENV transmission. 

Interpretation of the results from our study highlights the need for intersectoral collaboration, especially among health, urban planning, water, and agriculture/forestry sectors. Deforestation for city development, crops, and dams should be accompanied by the health and environment sectors to conduct the impact assessments, considering the anticipated detrimental impacts on public health. Moreover, these interconnected sectors should work together to define mitigation and adaptation strategies to reduce the effect of damage already inflicted. This is already part of the Constitution of the Kingdom of Thailand (2007), Article 672 [83,84], and also needs to be developed in Laos. 

Vulnerability index development depends on spatiotemporal socio-economic data and their quality. This study was limited by lack of spatial details since dengue case data used for validation were aggregated at the district and subdistrict levels and a finer scale georeferencing of individual patients’ households was not possible. The continuous spatio-temporal availability of climate and earth observation records from satellites improved the reliability of data for eco-environmental factors. However, fine scale temporal socioeconomic and demographic data remain challenging to access in endemic regions of South East Asia. A detailed annual socioeconomic and living conditions database was available for Thailand, but only for municipal and non-municipal areas without further division into subdistricts or villages. The performance of DVI_WADI_ could be improved by including detailed socioeconomic data for more time steps in Laos and Thailand at the village, subdistrict, and district levels.

## 5. Conclusions

Despite vector control and dengue prevention activities, dengue represents a significant disease burden in Laos and Thailand, indicating the need for further improvement of centralized national databases, understanding relevant environmental processes, and disseminating dengue-related knowledge in different socioeconomic settings. The vulnerability assessment using multiple approaches improves understanding of crucial dengue determinants and facilitates communication of complex interactions. The DVI_WADI_ approach in this study described changing conditions at a regional scale. It provided a visual tool that can support public health sectors and decision makers in allocating resources for interventions and prevention measures to reduce population susceptibility and enhance resilience. Further development of an interactive web-based platform of the WADI would be a practical step to facilitate communication of vulnerability trends to decision makers locally and globally.

## Figures and Tables

**Figure 1 ijerph-18-09421-f001:**
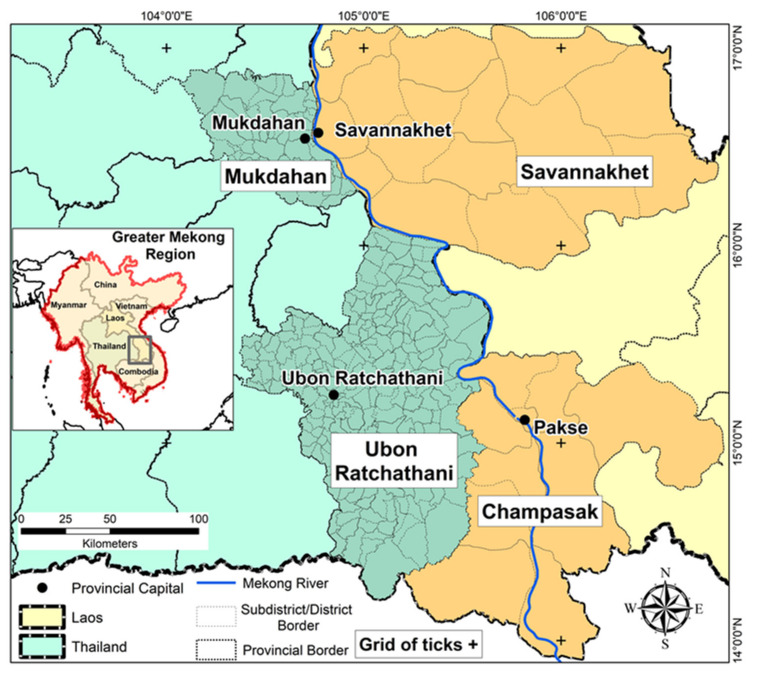
Study provinces in Central and Southern Laos and in Northeastern Thailand (+ grid ticks).

**Figure 2 ijerph-18-09421-f002:**
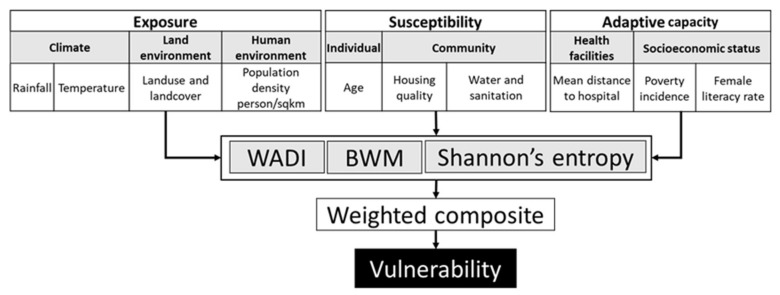
Conceptual framework for constructing the dengue vulnerability indices. Indicators of exposure, sensitivity, and adaptive capacity were combined using weights corresponding to the available evidence from the literature.

**Figure 3 ijerph-18-09421-f003:**
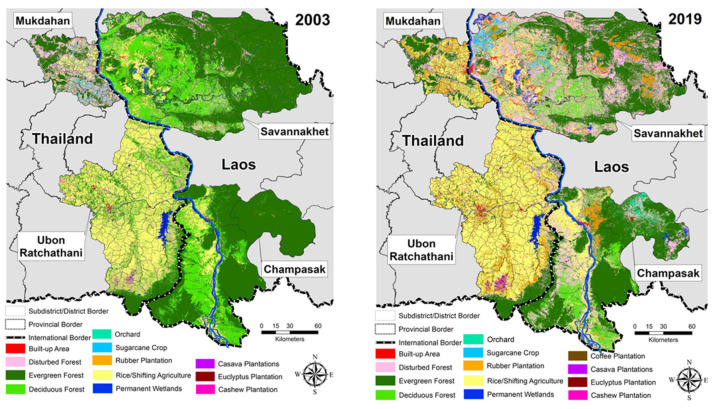
Land use and landcover in study provinces in Laos and Thailand from 2003 and 2019 mapped from Landsat satellite data.

**Figure 4 ijerph-18-09421-f004:**
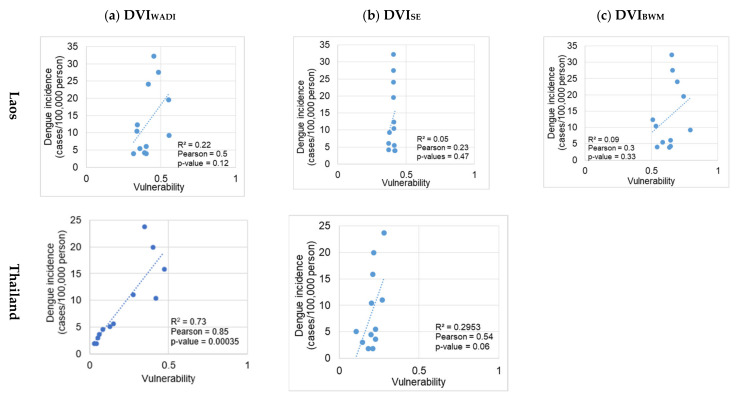
Scatter plot of average monthly dengue vulnerability indices and dengue incidence rate per 100,000 persons and (**a**) Dengue Vulnerability Index–Water Associated Disease Index (DVI_WADI_) and (**b**) Dengue Vulnerability Index–Shannon’s Entropy (DVI_SE_), (**c**) Dengue Vulnerability Index–Best-Worst Method (DVI_BWM_), in Laos (first row) and Thailand (second row).

**Figure 5 ijerph-18-09421-f005:**
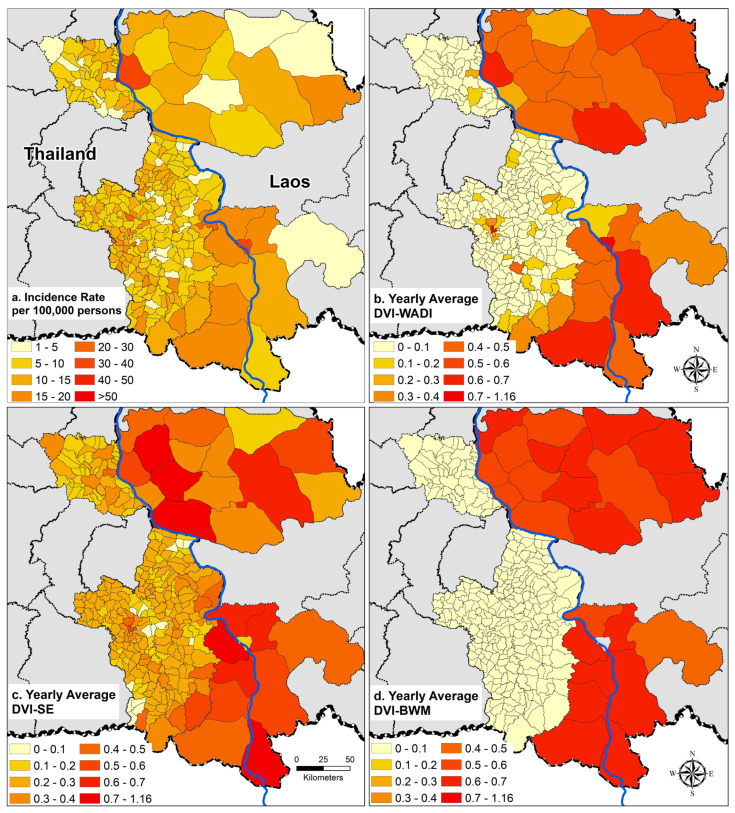
(**a**) Average dengue incidence rate per 100,000 persons and (**b**) Average Dengue Vulnerability Index–Water Associated Disease Index (DVI_WADI_) and (**c**) Average Dengue Vulnerability Index–Shannon’s Entropy (DVI_SE_), and (**d**) Average Dengue Vulnerability Index–Best-Worst Method (DVI_BWM_), in Laos and in Thailand from 2003–2019.

**Figure 6 ijerph-18-09421-f006:**
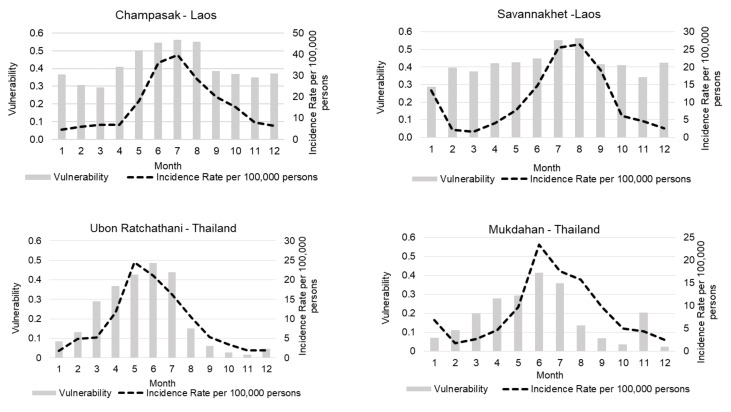
Average monthly vulnerability based on Water Associated Disease Index (DVI_WADI_) and incidence rate per 100,000 persons per month in selected provinces during 2003–2019.

**Figure 7 ijerph-18-09421-f007:**
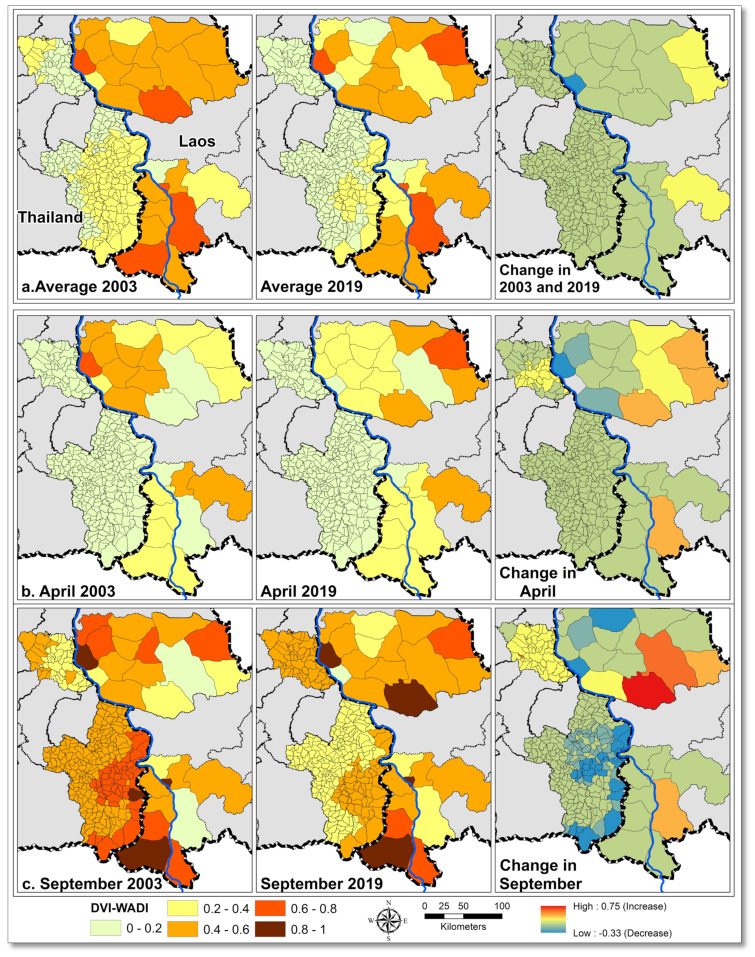
Dengue Vulnerability Index–Water Associated Disease Index (DVI_WADI_). (**a**) The upper panels show average DVI_WADI_ in 2003 (left), 2019 (middle), and change between 2003 and 2019 (right). (**b**) The middle panels show DVI_WADI_ in April 2003 (left), April 2019 (middle), and change between April 2003 and April 2019 (right). (**c**) The lower panels show DVI_WADI_ in September 2003 (left), September 2019 (middle), and change between September 2003 and September 2019 (right). DVI_WADI_ 0 = lowest and 1 = highest.

**Table 1 ijerph-18-09421-t001:** Determinants, indicators, and rationale for selection of data and their sources for vulnerability assessment.

Determinants	Indicators		Rationale for Selection	Data Source
Exposure	Climate	Mean Monthly Temperature (°C)	Higher temperatures favor vectorial reproduction such as the laying of eggs, egg hatching, and development of larva and pupa. The inverse relationship of temperature with the extrinsic incubation period of DENV in *Aedes* mosquitoes promotes viral transmission at higher temperatures (Carrington et al., 2013; Kuno, 1995).	Copernicus Climate Data Store [25,26] https://cds.climate.copernicus.eu/#!/home (accessed on 5 September 2021)
		Monthly Rainfall (mm)	Generation of aquatic habitats for oviposition [27].	
	Land environment	Forest, Plantations, cropland, or built-up land uses (%)	Human environments are favored by *Aedes aegypti* [28] and natural environments such as plantations by *Aedes albopictus* [29].	United States Geological Survey (USGS) [30], https://earthexplorer.usgs.gov (accessed on 5 September 2021)
	Human environment	Population density (person/km^2^.)	Availability of host reservoirs of virus, required for dengue transmission [31,32].	Total population [24] divided by Area
Determinants	Indicators		Rationale for selection	Data source
Susceptibility	Individual	Age under 15 years and greater than 60 years (%)	In Laos and Thailand, children 0–15 years have been reported to have a higher susceptibility to dengue than the adult population [33,34]. According to past 17 years of data used in this study, an increase in dengue cases is observed in the elderly population of an age greater than 60 years old (Appendix A: Age group distribution of reported dengue cases (DF, DHF, and DSS) in Savannakhet and Champasak provinces in Laos and Mukdahan and Ubon Ratchathani provinces in Thailand between 2003 and 2019).	National censuses Laos in 2005 and 2015 [23]. National annual socio-economic survey in Thailand in 2003–2019 [24].
	Community	Housing quality (%)	Houses with porous floors, unplastered walls, and bathrooms without tiles can cause increased indoor humidity, conducive to vector survival [35].	National censuses Laos in 2005 and 2015 [23]. Not available for Thailand at the subdistrict level
		Water and sanitation (%)	Unavailability of reliable piped water supply, water storage for drinking, and flush toilets increase oviposition sites [36,37,38].	
Determinants	Indicators		Rationale for selection	Data source
Adaptive Capacity		Female literacy rate (%)	Families with increased female education and literacy possess higher adaptive capacities [39].	National censuses Laos in 2005 and 2015 [23]. Not available for Thailand at subdistrict level
		Health facility in proximity/Mean distance to hospital (km)	Delay in medical attention of infant and child dengue patients; poor diagnosis and lack of appropriate care cause hospitalization and deaths [40,41].	National censuses Laos in 2005 and 2015 [23]. For Thailand mapped and calculated using Google Earth.
		Poverty incidence %	Households with low family income have low capacity to invest in health care [38].	National censuses Laos in 2005 and 2015 [23]. National annual socio-economic survey in Thailand in 2003–2019 [24].

**Table 2 ijerph-18-09421-t002:** Thresholds/Scores of indicators of determinants for Water Associated Disease Index [14] and Best-Worst Method. The dimensions of each indicator identify threshold values suitable for dengue transmission and given a score between 0 and 1 accordingly. A score of 1 indicates highest suitability for dengue transmission. Fractions of scores, e.g., 0.25 or 0.5, indicate lower transmission suitability. Zero indicates the least transmission suitability. Thresholds for rainfall and temperature were selected based on a retrospective analysis in study provinces and given a score of 1 for temperatures a suitable for dengue transmission and zero otherwise [43]. Scores for the indicators of each determinant were selected based on existing findings and from models of DENV transmission risk available in the literature [14,18,19].

Determinants	Indicators (Unit)	Dimensions	Thresholds and Scores
Exposure	Temperature (°C)	Monthly mean temperature, 1-month lag	24–29 °C: linear increase in exposure up to 1
			<24 °C or >29 °C: 0 Exposure
	Rainfall (mm)	Monthly cumulative rainfall, 1-month lag	<300 mm precipitation: linear increase in exposure up to 1
			>300: 0 Exposure
	Land use/Land cover (km^2^)	Built-up area	1
		Wetland area	0
		Rubber/cassava/cashew/coffee plantation area	0.5
		Forest area	0
		Disturbed forest area	0.25
		Rice and sugarcane crop area	0.25
	Population density (person/km^2^)	0–200	0.25
		200–400	0.5
		>400	1
Susceptibility	Age (%)	The proportion of population <15 to >60 years	1
		The proportion of population >15 to <60 years	0.5
	Living conditions/Housing quality (%)	The proportion of houses made of both concrete and wood	0.25
		The proportion of houses made of wood	0.5
		The proportion of houses made of bamboo and both bamboo and wood	1
	Toilet type (%)	The proportion of households with modern toilet	0
		The proportion of households with squat and pit toilet	1
Adaptive Capacity	Female literacy rate (%)	0–50	1
		50–100	0.25
	Mean distance to hospital (km)	0–5	0.25
		10–15	0.5
		>15	1
	Poverty incidence (%)	0–20	0.25
		20–40	0.5
		>40	1

**Table 3 ijerph-18-09421-t003:** Weight calculations of indicators using the Best-Worst Method (BWM).

**Exposure**	**Best to others: Temperature (°C)**		**Others to the Worst: Landcover (km^2^)**		**Weights**	**Consistency Index**
	Temperature (°C)	1	Temperature (°C)	2	0.39	0.07
	Rainfall (mm)	2	Rainfall (mm)	2	0.23	
	Landcover (km^2^)	2	Landcover (km^2^)	2	0.15	
	Population density (person/km^2^)	2	Population density (person/km^2^)	1	0.23	
**Susceptibility**	**Best to others: Population** **(Density person/km^2^)**		**Others to the Worst: Age**			0.04
	Living conditions	2	Living conditions	2	0.54	
	Toilet type	3	Toilet type	2	0.29	
	Age <15 and >60	4	Age <15 and >60	1	0.17	
**Adaptive capacity**	**Best to others: Literacy Rate (%)**		**Others to the Worst: Mean Distance to Hospital (km^2^)**			0.04
	Female literacy rate (%)	1	Female literacy rate (%)	3	0.54	
	Poverty incidence (%)	2	Poverty incidence (%)	2	0.29	
	Mean distance to hospital (km^2^)	3	Mean distance to hospital (km^2^)	1	0.17	
**Vulnerability**	**Best to others: Exposure**		**Others to the Worst: Adaptive Capacity**			0.04
	Exposure	1	Exposure	3	0.54	
	Susceptibility	2	Susceptibility	3	0.32	
	Adaptive capacity	3	Adaptive capacity	1	0.17	

**Table 4 ijerph-18-09421-t004:** Number of reported dengue cases and mean incidence rate (IR) and maximum incidence rate (IR) in two selected provinces in Laos and Thailand during 2003–2019. Outbreak years are those when maximum incidence rates exceeded 300 per 100,000 population in any month in any district. Months of highest IR are indicated in parentheses.

Year	Laos			Thailand		
	Observed Cases	Average IR	Maximum IR	Observed Cases	Average IR	Maximum IR
2003	5075	24.2	405.9 (October)	3709	13.5	340.4 (May) 553.8 (June)
2004	1452	6.6	349.3 (July)	1013	4.0	245.8
2005	2282	10.6	203.8	978	3.9	260.6
2006	1501	7.4	128.9	1021	3.9	277.8
2007	2146	10.9	132.4	1379	5.9	247.0
2008	3468	19.5	160.6	871	3.3	170.3
2009	1087	5.2	70.2	1059	4.5	533.0 (July)
2010	5391	26.2	345.4 (September)	2704	10.8	481.0 (June) 481.4 (July) 350.0 (August)
2011	572	3.0	73.5	1197	4.9	295.7
2012	1163	6.0	159.3	1135	4.3	160.8
2013	9294	44.0	487.0 (June) 578.9 (July) 312.0 (August)	4102	15.9	316.0 (March) 303.0 (May) 1404.2 (June) 637.0 (July) 405.0 (August)
2014	116	0.6	25.8	601	2.6	400.0 (June)
2015	210	1.0	98.9	5370	19.7	554.0 (July) 724.6 (August) 329.0 (September)
2016	1998	9.5	433.1 (June)	2644	10.1	402.9 (September)
2017	1688	9.0	148.4	1056	4.0	263.8
2018	1944	8.8	203.9	2284	8.8	418.7 (May) 395.0 (June)
2019	9465	33.0	336.0 (July) 575.3 (August) 411.0 (September)	8321	30.3	303.4 (April) 510.0 (May) 730.0 (June) 434.0 (July) 372.0 (August)
Mean	2874	12.6	–	2320	8.9	–

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
