# Peer review of "Development and Comparison of Dengue Vulnerability Indices Using GIS-Based Multi-Criteria Decision Analysis in Lao PDR and Thailand"

_ijerph, 2021, doi:10.3390/ijerph18179421_

Round 1
Reviewer 1 Report
This article needs a copy-edit. I am catching tons of typos, funny spacing, missing words, inappropriate pluralizing, etc. An English-language edit is also needed. I’m catching loads of mistakes in using (or not using) definite and indefinite articles and other confusing phrasing.
Did you define dengue incidence correctly on p. 4, line 169–170? Sounds like you are giving whole regional population over 100,000, but do you mean positive cases over 100,000?
After reading Table 2 alone, I did not understand the threshold values and what the indicated. The following paragraph explains it better, but I still don’t understand many of them. For some that are clearly explained (e.g., land-use type), I don’t think these are thresholds, but rather, scores, because this variable cannot be continuous as you are giving a numeric value to a categorical variable. Others still are not clear to me: how were housing-type scores determined? In the paragraphs where you are explaining these values, you discuss changes from 2003 and 2019. Is that change being factored into these values, or was that extraneous information that should be placed elsewhere in the manuscript?
Much of the notation for the weighting equations is hard to understand. Given the great number of typos throughout the manuscript, I do not know if some of my confusion is because some of the mathematical notations are not explained properly or if they are typos. Equations and weighting methods need better evaluation after a copyedit.
If significant data sources were missing for Thailand, how valuable are the interpretations of these results? It’s striking to me that the strongest R2 values (Fig 4) came from the comparisons with the least data. While I do not think the interpretation of the results is incorrect, I think it needs to be discussed much more cautiously.
Author Response
Dear reviewer,
Thank you for your comments and suggestions. Please find our responses below to your queries.
Q1. This article needs a copy-edit. I am catching tons of typos, funny spacing, missing words, inappropriate pluralizing, etc. An English-language edit is also needed. I’m catching loads of mistakes in using (or not using) definite and indefinite articles and other confusing phrasing.
- Thank you for the kind advice and accept my apology for these incorrect spacing and typos. We have corrected all the spacing issues, rechecked, and correct typos, and grammatical mistakes.
Q2. Did you define dengue incidence correctly on p. 4, lines 169–170? Sounds like you are giving the whole regional population over 100,000, but do you mean positive cases over 100,000?
- Dengue incidence was calculated for each spatial unit by dividing the positive cases with its total population and multiplied by 100,000 (Page 4, line142–144).
Q3. After reading Table 2 alone, I did not understand the threshold values and what they indicated. The following paragraph explains it better, but I still don’t understand many of them. For some that are clearly explained (e.g., land-use type), I don’t think these are thresholds, but rather, scores, because this variable cannot be continuous as you are giving a numeric value to a categorical variable.
A.1. These are the scores for all the indicators. However, the continuous variable’s thresholds were identified from the literature and given scores.
A.2. An updated caption for Table 2 is added on page 9, lines 185-192, and also written below.
Table 2: Thresholds/Scores of indicators of determinants for Water Associated Disease Index [14] and Best–Worst Method. The dimensions of each indicator identify threshold values suitable for dengue transmission and given a score between 0 and 1 accordingly. A score of 1 indicates the highest suitability for dengue transmission. Fractions of scores, e.g., 0.25 or 0.5, indicate lower transmission suitability. Zero indicates the least transmission suitability. Thresholds for rainfall and temperature were selected based on a retrospective analysis in study provinces and given a score of 1 for temperatures suitable for dengue transmission and zero otherwise [42]. Scores for the indicators of each determinant were selected based on existing findings and from models of DENV transmission risk available in the literature[14,18,19].
Q4. Others still are not clear to me: how were housing-type scores determined?
- Percentage of houses in three housing categories were provided as one whole figure for each district in the 2005 census and for 2015, the data were available at the village level. We calculated percentage housing categories for each district—the districts assigned with the most abundant housing category value. Districts with most houses constructed with concrete and wood were given the lowest score of 0.25, a score for houses with wood 0.5, and houses with wood and bamboo were assigned the highest score of 1. (Page 11, Line 240-246)
Q5. In the paragraphs where you are explaining these values, you discuss changes from 2003 and 2019. Is that change being factored into these values, or was that extraneous information that should be placed elsewhere in the manuscript?
- Yes, changes over the studied period are factored into these values (Lines 194-195, 224-225, and 253-255).
Q6. Much of the notation for the weighting equations is hard to understand. Given the great number of typos throughout the manuscript, I do not know if some of my confusion is because some of the mathematical notations are not explained properly or if they are typos.
- Sorry for the confusion. They were not typos. We have updated all mathematical equations so that all notations should now be correct.
Q7. Equations and weighting methods need better evaluation after a copyedit.
- We are not sure if we fully understand what the reviewer meant by better evaluation of equations and weighting methods. We have rechecked all the weighting methods and their equations after the copyedit. We hope that the revised version is understandable
Q8. If significant data sources were missing for Thailand, how valuable are the interpretations of these results? It’s striking to me that the strongest R2 values (Fig 4) came from the comparisons with the least data. While I do not think the interpretation of the results is incorrect, I think it needs to be discussed much more cautiously.
- Thank you for pointing this out. Please find our response below and in the manuscript pages 26-27, lines 643-646, and 658-661.
For Thailand: On average, exposure contributed 56% to the DVIWADI score in Thailand and only 11% in Laos. The relatively higher contribution of exposure to dengue vulnerability in Thailand could be a reason for the overall high correlation between DVIWADI and dengue incidence (R2 =0.73 and r=0.85), even if data for susceptibility and adaptive capacity determinants were limited (Figure 4)
For Laos: The relatively higher contribution of susceptibility and limited data (censuses in 2005 and 2015) in Laos might be the reason for an overall lower correlation between DVIWADI and dengue incidence (R2 =0.22 and r=0.5) (Figure 4).
Reviewer 2 Report
In-depth study of Dengue Fever Vulnerability Index using GIS. GIS has very important application value in the field of public health, and its origin is London cholera.
After careful reading, I think the following questions should be further revised:
1) Relevant work should be strengthened, especially the research work of other researchers in recent 5 years on the Web of Science website;
2) The title of Figure 1 seems to be messed up, and the latitude and longitude label does not indicate the specific location, which is not conducive to the understanding of the figure;
3) Can you add some more detailed data or more accurate data for experiment?
4) The legend in Figure 3 can be more clear;
5) Limitaions and Recommendations can be included in Discussion or Conclusion?
6) The conclusion seems to be a little thin and can give a better conclusion?
7) Some minor mistakes in writing can be avoided.
Author Response
Dear reviewer,
Thank you for your comments and suggestions for ours manuscript. Please find our responses to your queries below.
In-depth study of Dengue Fever Vulnerability Index using GIS. GIS has very important application value in the field of public health, and its origin is London cholera.
After careful reading, I think the following questions should be further revised:
Q1. Relevant work should be strengthened, especially the research work of other researchers in recent 5 years on the Web of Science website.
- We already reviewed the latest literature during the last 5 years. Here I have listed a few from my reference list.
- Ali, S.A.; Ahmad, A. Mapping of mosquito-borne diseases in Kolkata Municipal Corporation using GIS and AHP based decision-making approach. Inf. Res. 2019, 27, 351–372, doi:10.1007/s41324-019-00242-8.
- Jain, R.; Sontisirikit, S.; Iamsirithaworn, S.; Prendinger, H. Prediction of dengue outbreaks based on disease surveillance, meteorological and socio-economic data. BMC Infect. Dis. 2019, 19, doi:10.1186/s12879-019-3874-x.
- Bavia, L.; Melanda, F.N.; de Arruda, T.B.; Mosimann, A.L.P.; Silveira, G.F.; Aoki, M.N.; Kuczera, D.; Sarzi, M. Lo; Junior, W.L.C.; Conchon-Costa, I.; et al. Epidemiological study on dengue in southern Brazil under the perspective of climate and poverty. Rep. 2020, 10, 1–16, doi:10.1038/s41598-020-58542-1.
- Pham, N.T.T.; Nguyen, C.T.; Vu, H.H. Assessing and modelling vulnerability to dengue in the Mekong Delta of Vietnam by geospatial and time-series approaches. Res. 2020, 186, 109545, doi:10.1016/j.envres.2020.109545.
- Henry, S.; Mendonça, F. de A. Past, present, and future vulnerability to dengue in jamaica: A spatial analysis of monthly variations. J. Environ. Res. Public Health 2020, 17, doi:10.3390/ijerph17093156.
- Telle, O.; Nikolay, B.; Kumar, V.; Benkimoun, S.; Pal, R.; Nagpal, B.; Paul, R.E. Social and environmental risk factors for dengue in Delhi city: A retrospective study. PLoS Negl. Trop. Dis. 2021, 15, e0009024, doi:10.1371/journal.pntd.0009024.
- Tsheten, T.; Clements, A.C.A.; Gray, D.J.; Wangdi, K. Dengue risk assessment using multicriteria decision analysis: A case study of Bhutan. PLoS Negl. Trop. Dis. 2021, 15, e0009021, doi:10.1371/journal.pntd.0009021.
Q2. The title of Figure 1 seems to be messed up, and the latitude and longitude label does not indicate the specific location, which is not conducive to the understanding of the figure.
- Figure title fixed and grid ticks added to the map to understand the latitude and longitude of the specific location. (Figure 1, Page 3)
Q3. Can you add some more detailed data or more accurate data for experiment?
- We do not understand what kind of data the reviewer thinks are missing, and which would be more accurate. We have collected detailed environmental data from 2003 – 2019, including land use and land cover and climatic variables (rainfall and temperatures). Two censuses’ data for Laos (2005 and 2015) was used to understand and calculate the weights for susceptibility and adaptive capacity determinants of vulnerability. For Thailand, detailed census data were not available at the subdistricts level. We contacted the statistics department, and they provided data in two categories of municipal and non-municipal areas of each province, not for each subdistrict. Data limitations are explained on page 27 in lines 699 – 710.
Q4. The legend in Figure 3 can be more clear
- All figures are replaced in the manuscript with high-resolution ones.
Q5. Limitations and Recommendations can be included in Discussion or Conclusion?
- The Limitations and Recommendations are now added to the discussion.
Q6. The conclusion seems to be a little thin and can give a better conclusion?
- The conclusion is updated.
Q7. Some minor mistakes in writing can be avoided.
- Sorry for these minor mistakes. Thank you for your kind advice. I have corrected all the spacing issues, rechecked and correct typos, and grammatical mistakes
Round 2
Reviewer 2 Report
All my concerns have been answered.